# RNase A Domain-Swapped Dimers Produced Through Different Methods: Structure–Catalytic Properties and Antitumor Activity

**DOI:** 10.3390/life11020168

**Published:** 2021-02-21

**Authors:** Riccardo Montioli, Rachele Campagnari, Sabrina Fasoli, Andrea Fagagnini, Andra Caloiu, Marcello Smania, Marta Menegazzi, Giovanni Gotte

**Affiliations:** 1Department of Neuroscience, Biomedicine, and Movement Sciences, Biological Chemistry Section, University of Verona, Strada Le Grazie, 8, I-37134 Verona, Italy; riccardo.montioli@univr.it (R.M.); rachele.campagnari@univr.it (R.C.); sabrina.fasoli@univr.it (S.F.); andrea.fagagnini@univr.it (A.F.); marcello.smania92@studenti.univr.it (M.S.); 2Department of Microbiology and Virology, Wexham Park Hospital, Wexham Road, Slough SL24HL, Berkshire, UK; andracaloiu@nhs.com

**Keywords:** RNase A, RNase A dimers, 3D domain swapping, enzymatic activity, cytotoxic activity, melanoma cells

## Abstract

Upon oligomerization, RNase A can acquire important properties, such as cytotoxicity against leukemic cells. When lyophilized from 40% acetic acid solutions, the enzyme self-associates through the so-called three-dimensional domain swapping (3D-DS) mechanism involving both N- and/or C-terminals. The same species are formed if the enzyme is subjected to thermal incubation in various solvents, especially in 40% ethanol. We evaluated here if significant structural modifications might occur in RNase A N- or C-swapped dimers and/or in the residual monomer(s), as a function of the oligomerization protocol applied. We detected that the monomer activity vs. ss-RNA was partly affected by both protocols, although the protein does not suffer spectroscopic alterations. Instead, the two N-swapped dimers showed differences in the fluorescence emission spectra but almost identical enzymatic activities, while the C-swapped dimers displayed slightly different activities vs. both ss- or ds-RNA substrates together with not negligible fluorescence emission alterations within each other. Besides these results, we also discuss the reasons justifying the different relative enzymatic activities displayed by the N-dimers and C-dimers. Last, similarly with data previously registered in a mouse model, we found that both dimeric species significantly decrease human melanoma A375 cell viability, while only N-dimers reduce human melanoma MeWo cell growth.

## 1. Introduction

Natively monomeric 13.7 kDa ribonuclease A (RNase A, EC 3.1.27.5), the proto-type of the secretory “pancreatic-type”(pt)-RNase superfamily [1] can self-associate [2] upon lyophilization from 30–50% aqueous acetic acid (HAc) solutions [3]. Consequently, the enzyme produces non-covalently linked dimers, trimers, and larger oligomers [2,4,5,6] through the so-called three dimensional domain swapping (3D-DS) mechanism [7] involving its N- and/or C-terminal domains [2,4]. Some mutants, like S80R, or variants affecting the N- and/or C-termini polarity increase the RNase A 3D-DS oligomerization tendency [8,9]. Likewise, the introduction of Cys31 and 32 plus other mutations making the enzyme similar to the natively domain-swapped dimeric BS-RNase [10], augment the RNase A propensity to self-associate [2,11,12]. Oligomerization occurs also when highly concentrated RNase A solutions are subjected to thermal incubation in various solvents, such as 40% aqueous ethanol (EtOH), obtaining different relative oligomers amounts as a function of the solvent used and the temperature experimented [13].

The structures of both RNase A N- or C-swapped dimers (N-dimer, or N_D_; C-dimer, or C_D_, respectively), as well as of a cyclic C-swapped trimer (C_T_) formed through acidic lyophilization have been solved [14,15,16], while models have been proposed for higher-order domain-swapped oligomers [4,5,6,16]. Instead, no data are so far available on the 3D structures of the dimers obtained through thermal incubations. Interestingly, the 3D-DS mechanism is shared by some amyloidogenic proteins, such as human prion protein or cystatin C, both forming domain-swapped dimeric or oligomeric precursors of their amyloid fibrils involved in severe deposition diseases [17,18]. The toxicity of these proteins is generally ascribable to their oligomers [19,20], and these species become toxic, or not, as a function of the protein aggregation pathway(s) followed [20,21], or of the environmental pH [22,23]. However, wt-RNase A cannot undergo fibrillation, although displaying many aggregation-prone sequences [24]. Conversely, the limited RNase A self-association leads to the formation of active oligomers that are in turn able to enter tumor cells and digest intracellular RNAs and therefore driving malignant cells to damage and death [25]. This antitumor activity generally increases with size and basic charge exposure of the oligomers [2], as well as with their higher ability to evade the intracellular ribonuclease inhibitor (RI) with respect of the monomer [11,26]. Furthermore, animal models had demonstrated that the blood half-life of a RNase A covalently formeddimer at least doubles the one of the monomeric form, thus reflecting the protein renal clearance rate [27]. In this way, a longer time-contact of RNase A dimers with malignant cell surfaces and a consequent efficient cellular uptake could occur, thus favoring in vivo cytotoxicity.

In this work, we evaluated if two different protocols useful to induce non covalent RNase A oligomerization through 3D-DS may affect the structural and/or functional properties of the residual monomer and/or the relative dimers. To do so, we compared the features of both RNase A dimers, i.e., N_D_ and C_D_ [14,15], and the residual monomer (M) recovered after: (i) RNase A lyophilization from 40% aqueous HAc solutions [3], and here called X_-H_, or, in detail, M_-H_, N_D-H_, C_D-H_, or (ii) 60 °C incubation of highly concentrated RNase A dissolved in 40% aqueous EtOH, i.e., the solvent providing the highest oligomers yield [13]. We call here the resulting species X_-Et_, or specifically M_-Et_, N_D-Et_ and C_D-Et_, respectively. Interestingly, we report for the first time a comparison between the kinetic parameters of the mentioned X_-H_ and X_-Et_ species measured vs. yeast RNA. We propose, in addition, hypotheses to explain why RNase A C-dimers exert a lower enzymatic activity vs. ss-RNA with respect to N-dimers [28]. Finally, the anti-tumor activity of all dimers deriving from both RNase A incubation methods has been investigated in two human melanoma cell lines, in comparison with the respective monomers. The data obtained are discussed and compared with previous results registered in a melanoma mouse model [29].

## 2. Materials and Methods

### 2.1. Materials

RNase A type-XIIA, polyadenylic:polyuridylic acid (poly(A):poly(U), divinylsulfone (DVS), HAc and EtOH were from Sigma, Milan, Italy. Acetonitrile (ACN), trichloroacetic acid (TCA), and trifluoroacetic acid (TFA) were from Merck-Millipore, Milan, Italy. Yeast RNA was from Boehringer, Germany. All other chemicals were of the highest purity available.

### 2.2. RNase A Oligomerization

#### 2.2.1. Lyophilization from 40% Acetic Acid Solutions

The oligomerization of native, monomeric RNase A (M) was induced by lyophilizing its 20 to 50 mg/mL solutions in 40% aqueous HAc [3]. The powder was re-dissolved in 50 mM inorganic phosphate buffer (NaPi), pH 6.7, at about 10 mg/mL concentration, and kept into an ice bath until performing chromatographic purification and additional experiments [2,30].

#### 2.2.2. Thermally Induced Oligomerization

RNase A oligomers were obtained also by incubating 180–200 mg/mL protein solutions at 60 °C for 2 h in 40% aqueous EtOH [13]. Thereafter, 0.5 mL of 50 mM pH 6.7 NaPi buffer previously heated to 60 °C was added to stop the incubation [13]. Once spontaneously cooled to room temperature (RT), the mixture was brought into an ice bath and stored until performing chromatographic purification and additional experiments [13].

### 2.3. Mass Spectrometry of RNase A Monomers

The molecular weight (MW) of native RNase A and of each monomer recovered upon incubations, i.e., M_-H_ and M_-Et_, was determined by Mass Spectrometry (MS). Each sample was suspended with an aqueous ACN solution (30:70 (v/v) ACN: 0.1% aqueous TFA), and incubated for 30 min at RT. The solution was 1:1 (v/v) mixed with the sinapinic acid (trans-3,5-dimethoxy-4-hydroxycinnamic acid) matrix (10 mg/mL in ACN:H_2_O 1:1 containing 0.1% TFA). Then, 1 µL of the sample/matrix solution was spotted in triplicate onto a Ground Steel MALDI target plate (Bruker Daltonics, Billerica, MA, USA) allowed to dry at RT. MS analysis was performed on an Ultraflextreme MALDI-TOF/TOF instrument (Bruker Daltonics) of the “Centro Piattaforme Tecnologiche” (CPT) of the University of Verona. Spectra were collected in the positive linear mode with a 5000 to 20,000 m/z range. Mass calibration was performed with a standard mixture of myoglobin, cytochrome C, ubiquitin-I and insulin. Instrument settings: ion source 1 and 2: 19.93 kV and 18.77 kV, respectively; lens: 8.57 kV; delay time: 104 ns with 800 scale factor; acceleration voltage: 20 kV; number of shots: 2000, in six different positions for one spectrum. All data were analyzed with the Flex Analysis Software (Bruker Daltonics).

### 2.4. Chromatographic Purification of RNase A Monomers and Dimers

Samples recovered upon subjecting RNase A to one or the other oligomerization protocol [3,13] were purified with a Source 15S HR 10/10 cation-exchange column connected to an ÄKTA FPLC system (GE Healthcare, Milan, Italy) and equilibrated with 70 mM NaPi buffer, pH 6.7. After eluting the monomer, NaPi concentration was increased to 90 mM to recover RNase A N_D_, and then NaPi concentration was increased to 200 mM, by applying a 1 h linear gradient, to separate the C-dimer (C_D_) from the larger oligomers [28]. Additional SEC analyses have been performed with a Superdex 75 Increase HR 10/300 column (GE-Healthcare), attached to the same ÄKTA FPLC system: the column was equilibrated with 0.2 M NaPi, pH 6.7, and elution performed at 0.2 mL/min flow rate [2].

Each N_D-H_ or N_D-Et_ [14], and C_D-H_ or C_D-Et_ [15] species was separately collected in parallel with the corresponding monomers, M_-H_ and M_-Et_. The protein samples concentration was increased to values detectable in a Jasco V-650 spectrophotometer (RNase A ε^1%^_280_ = 7.3 [31]), by using Amicon-3 ultrafilters (3 kDa cutoff, Millipore-Sigma, Milan, Italy). With the same ultrafilters, the samples NaPi concentration was lowered to 10, 20, or 50 mM depending on the subsequent uses planned. Then, samples were kept at 4 °C to slow down their dissociation to monomer.

Possible protein deamidation events induced by the incubations were investigated, beyond MS analyses, by injecting 0.5 mg of M_-H_, or M_-Et_ onto the same Source 15S HR 10/10 column equilibrated with 60 mM NaPi, pH 6.7, and eluting them with no gradient applied [32].

### 2.5. Cathodic Non Denaturing PAGE

12.5–15% acrylamide cathodic PAGE of the RNase A samples were performed under non-denaturing conditions by using a Mini-Protean device (BioRad, Hercules, CA, USA) placed into an ice bath. 4× 0.08%(m/m) methyl green/20%(m/m) glycerol loading buffer was added to the samples, and protein migration was carried out at 180–200 V for 90 min, with the gel immersed in the 0.35% β-alanine/HAc pH 4.0, buffer [30]. Gel bands were stained with 0.1% Coomassie Brilliant Blue, then destained in a 10% HAc/20% EtOH aqueous solution.

### 2.6. Divinylsulfone (DVS) Cross-Linking Reaction of The Dimers

Divinylsulfone (DVS) is a bi-functional reagent that can covalently link the two His residues of the pt-RNases active site, one belonging one to the N-terminus, the other to the C-terminal end [14,15,16,33]. Therefore, if dimerization occurs through the 3D-DS of one or both termini, the cross-linking would provide a covalent dimer resisting to the dissociation induced by SDS-PAGE [33].

All RNase A dimers were brought to pH 5.0 in 0.10 M NaAc/HAc buffer at a concentration of about 0.5 mg/mL, and incubated with 10% V/V DVS as a 1000:1 molar ratio excess with respect to each protein subunit [33]. In order to stop the reaction and analyze its time-course with SDS-PAGE, β-mercaptoethanol was added to aliquots of the reaction mixture at different times up to 72 h.

### 2.7. Circular Dichroism Spectroscopy and Thermal Denaturation Analyses

Far-UV CD spectra of 0.30 mg/mL RNase A monomers solutions and 0.10 mg/mL dimers, dissolved in 10 mM NaPi, pH 6.7, were measured between 190 and 240 nm. The near-UV-CD spectra of the 0.45–0.50 mg/mL monomers were instead recorded from 240 to 340 nm. All measures were performed with a Jasco J-710 spectropolarimeter (Jasco Europe, Cremella (LC), Italy). All spectra reported resulted from 6 scans with 1 nm bandwidth and 20 nm/min scan speed, normalized against concentration after subtracting the spectrum of NaPi buffer.

The thermal denaturation profiles of all RNase A monomers and dimers dissolved in 20 mM NaPi were followed by measuring the 218 nm far-UV CD signal variation with the same Jasco J-710 spectropolarimeter, equipped with a Peltier thermostatic device. The temperature of each sample was increased by 1.5 °C/min from 20 to 90 °C, and the 2 nm bandwidth signal recorded every 0.5 °C [32]. Each T_m_ value was derived from the 50% far-UV CD signal.

The thermal unfolding reversibility was investigated by incubating native RNase A at 90 °C for times ranging from 5 to 240 min, and then cooling the sample down to 25 °C. Each relative near-UV CD spectrum was measured and compared with the one of the untreated, native enzyme, M. Then, aliquots were collected after different times during the 90 °C incubation and analyzed with cathodic PAGE to detect if a possible concomitant RNase A deamidation occurred [32,34].

### 2.8. Urea Denaturation Analyses of RNase A Dimers

The RNase A N- and C-dimers (N_D-H/Et_, C_D-H/Et_) dissolved in NaPi 20 mM, pH 6.7, were incubated at 25 °C with different urea concentrations ranging from 0 to 6 M. The final concentration of the dimers ranged from 0.5 to 0.7 mg/mL. After 2 h incubation, samples containing about 5 to 7 µg dimers were analyzed with a 12.5% acrylamide cathodic PAGE performed under non denaturing conditions [30] to detect the possible dissociation to monomer.

### 2.9. Intrinsic Fluorescence of RNase A Monomers and Dimers

The intrinsic fluorescence emission spectra of 0.1 mg/mL solutions of native RNase A, M, and all X_-H_ and X_-Et_ RNase A species [3,13] were registered from 290 to 500 nm, upon 280 nm excitation, with a 5 nm bandwidth. Measures were performed in triplicate with a 1 cm path-length quartz cuvette at 25 °C, using a Jasco FP-750 spectrofluorimeter (Jasco Europe, Cremella (LC), Italy).

### 2.10. Enzymatic Activity

#### 2.10.1. Single-Stranded RNA (ss-RNA)

The enzymatic activities of RNase A monomers and dimers [3,13] were tested vs. yeast ss-RNA as a substrate following the protocol described by Kunitz [35]. Assays were performed in triplicate per each RNase A species by monitoring the decrease of the 300 nm substrate absorbance (Abs_300_) in 300 μL of 0.10 M sodium acetate/HAc buffer solutions, pH 5.0, with a substrate concentration varying from 0.02 to 0.80 mg/mL [35]. The final enzyme concentration was 0.80 ng/μL for the monomers, 1.50 ng/μL for both dimers. Blank and negative controls were (i) the buffer and (ii) the Abs_300_ of the enzyme-free yeast RNA, respectively. The absorbance was followed in the linearity range and the catalytic activity measured as ΔAbs_300_/min/µg enzyme [35].

#### 2.10.2. Double-Stranded RNA (ds-RNA)

The enzymatic activity vs. ds-RNA was spectrophotometrically measured in quadruplicate at 260 nm, in the linearity range, and with the poly(A):poly(U) substrate diluted to concentrations ranging from 0.02 to 0.20 mg/mL in 300 μL of 0.150 M NaCl/0.015 M sodium citrate buffer, pH 7.0 [36]. We could not test higher substrate concentrations because of the concomitant excessive spectrophotometric error incoming. In the assays, 10 μg of each 1 mg/mL RNase A monomer solution were added, while 4 μg of N_D_ or 2 μg of C_D_, as 0.4–0.5 mg/mL solutions [5,28]. All RNase A species were dissolved in 20 mM NaPi, pH 6.7. Blank and negative controls were (i) the buffer and (ii) the enzyme-free poly(A):poly(U) 260 nm absorbance, respectively. The specific catalytic activity was determined as ΔAbs_260_/min/μg enzyme [36], while the specific activity/mg substrate is the slope of the curve.

### 2.11. Malignant Melanoma Cell Culture and Viability Assays

The human MeWo (HTB-65™) and A375 (CRL-1619™) melanoma cell lines (ATCC^®^, Manassas, VA, USA) were cultured at 37 °C, 5% CO_2_, using the Roswell Park Memorial Institute 1640 medium (RPMI, Gibco, Ref.21875, Thermo Scientific, Waltham, MA, USA) for MeWo cells, or Dulbecco’s Eagle’s Medium (DMEM, Gibco, Ref.61965, Thermo Scientific, Waltham, MA, USA) for A375 cells, in humidified atmosphere. The culture medium was supplemented with 10% Fetal Bovine Serum (FBS, Gibco, Thermo Scientific, Waltham, MA, USA) plus 1% Antibiotic Antimycotic solution (Gibco, Thermo Scientific, Waltham, MA, USA).

MeWo or A375 cells (5.0 × or 2.9 × 10^3^ cells/well, respectively) were seeded in 96-well plates. After 24 h, cells were treated with 100 µg/mL RNase A monomers (M-_H_ or M-_Et_) or with 25, 50, 100 µg/mL N_D-H_, or N_D-Et_, or C_D-H_, or C_D-Et_. After 72 h incubation, cells were fixed by adding 25 µL/well of 50% (W/V) TCA directly into the culture medium. Plates were incubated at 4 °C for 1 h, washed four times with dd-H_2_O and dried at RT. Staining was performed by adding 50 µL/well of 0.04% (W/V) sulforhodamine B sodium salt solution (SRB, Sigma-Aldrich, Milan, Italy). After 1 h incubation at RT, plates were rinsed with 1% HAc and air-dried. SRB was solubilized in 10 mM Tris solution, at pH 10.5, and Abs_492_ measured in the Tecan NanoQuant Infinite M200-Pro plate reader (Tecan Group Ltd., Männedorf, Switzerland) to determine the cell viability percentage compared to the control. Four to five indipendent experments were carried out with N_D-H_ or N_D-Et_, as well as with C_D-H_ or C_D-Et_, each derived from three different preparations. Six replicates were performed for each condition related to different concentration of the RNase A species.

### 2.12. Statistics

All the results are reported as a mean value ± SD. Unless otherwise noted, *p* values were determined using unpaired, two-tailed Student’s *t* test, with one asterisk * if *P* < 0.05, or two asterisks ** if *P* < 0.01. For each type of experiment, a minimum of four independent biological replicates were performed.

## 3. Results

### 3.1. RNase A Oligomerization and Purification of The Oligomers

All X_-H_ and X_-Et_ species obtained upon RNase A lyophilization from 40% HAc [3], or upon incubating 200 mg/mL protein solutions in 40% EtOH at 60 °C [13], were purified with a cation-exchange Source 15S HR 10/10 column, as described in the Materials & Methods Section [28]. The Appendix A report the elution profiles showing the different relative amounts of the dimers, that are known to depend on the incubation type underwent [13,28]. The two RNase A oligomerization mixtures have been analyzed also with a SEC Superdex 75 Increase HR 10/300 column, providing comparable results (Appendix A).

### 3.2. No Deamidation Onset Is Detectable in RNase A after Suffering Both Incubation Types

The MS of the monomers revealed a 2 Da MW excess for M_-H_ with respect to native RNase A (M) (Appendix A), a result within or close the uncertainty of the technique. We then used cation exchange chromatography to test if at least Asn67, the RNase A residue most sensitive to deamidation [37], suffered this modification [32], since we previously verified that this chromatographic approach can detect the deamidation of this single residue [32,37]. In this present work, instead, no differences, hence no deamidation events have been detected to be suffered by both M_-H_, and M_-Et_ chromatographic profiles with respect to native M (Appendix A).

### 3.3. Electrophoretic Mobility of the RNase A Species under Non Denaturing Conditions

All X_-H_ and X_-Et_ RNase A species were analyzed also with non-denaturing cathodic PAGE, and no differences emerged within bands relative to M_-H_, M_-Et_ and native M, as well as within each N_D-H_/N_D-Et_ and C_D-H_/C_D-Et_ pair (Figure 1A).

The absence of altered electrophoretic mobility, firstly ascribable to the conversion of Asn to a negatively charged Asp, or IsoAsp [32], confirms again that both treatments did not cause here any deamidation event. This is in line with cation exchange chromatography profiles visible in the Appendix A. Instead, the different mobility of the two N_D_ with respect to C_D_ originates from the compromise between the lower compactness and the higher basic charge exposure of the latter ones [28,30].

### 3.4. Divinyl-Sulfone (DVS) Cross-Linking States that RNase A Dimerizes through 3D-DS Also upon Thermal Incubation

The high similarity of the RNase A species obtained upon applying the two mentioned protocols had already been reported [13]. However, the 3D-DS mechanism was certified to be followed only by RNase A oligomers produced upon acidic lyophilization [14,15,16], while it was not proven yet for the thermally induced self-association.

Therefore, we incubated both N_D-H_/N_D-Et_ and C_D-H_/C_D-Et_ couples with the DVS bi-functional reagent to covalently link the two active site His residues, respectively located one in the N-terminus and the other in the C-terminus of the enzyme [5,33,38]. Each dimer was brought to pH 5.0 at about 0.6 mg/mL concentration and incubated with 1000 molar DVS excess over each protein subunit [33]. Aliquots of the reaction mixture were treated, at different times up to 72 h, with β-mercaptoethanol [33].

Figure 1B shows that, besides N_D-H_ and C_D-H_, also N_D-Et_ and C_D-Et_ are dimers resistant to SDS-PAGE dissociation. This indicates that DVS actually links the N- and C-terminal His residues of the two dimers subunits, and certifies that RNase A oligomerizes through the 3D-DS mechanism also upon thermal incubation(s). Incidentally, the monomer band splitted during the reaction time-course. This is known to be due to a proteolytic side-effect induced by DVS on the five C-terminal RNase A residues located after H119 [33]. Again, the incompleteness of dimer covalent stabilization, deducible by the presence of the monomer after 72 h reaction, can be ascribed to a partial dimers’ dissociation that occurs during the reaction requiring pH = 5.0 [5,33,38].

### 3.5. CD-Spectra and T_m_ Values, of the RNase A Species Are Not Affected by the Enzyme Incubations

The far-UV CD spectra of all RNase A species dissolved in 10 mM NaPi buffer, pH 6.7, were recorded between 190 and 240 nm. The three monomers spectra resulted almost identical (Figure 2A, upper panel), and only negligible relative differences were visible between the spectra of HAc- or EtOH-derived dimers (middle and lower panels). On the basis of these observations, none of the two incubation methods significantly altered the secondary structure composition of all RNase A species. Then, also the 240–340 nm near-UV CD spectra of RNase A monomers did not indicate relevant tertiary structure differences within M, M_-H_ and M_-Et_ (Figure 2B). The high amount of protein required did not allow us to measure the near-UV-CD spectra of all N- or C-dimers.

The thermal denaturation of each X_-H_ and X_-Et_ species dissolved in 10 mM NaPi pH 6.7 was followed by monitoring their 218 nm far-UV CD signal from 25 to 90 °C. The denaturation curves were analyzed with a two-state model and the melting temperature of all RNase A monomeric and dimeric species fell around 65 °C, as reported in Table 1. 

We also investigated the RNase A denaturation reversibility after heating up to 90 °C. The relative far UV-CD spectra reported in Figure 2C revealed the enzyme is totally renatured if it is cooled-down within the first ten minutes 90 °C incubation. Instead, it is definitely affected, showing two isosbestic points around 207 nm and 232 nm, if kept at 90 °C for more than ten minutes [39]. This was confirmed by cathodic PAGE, as reported in Figure 2D: non denaturing electrophoresis preserves the net charge of the samples, and in our case the mobility of the bands indicates that RNase A remained unaltered with respect to native monomer (Figure 2D, lanes 1 and 9) only within the first ten min 90 °C incubation (lanes 2,3). Afterwards, (lanes 4–8), bands display a lower mobility than native RNase A, indicating the presence of a higher negative charge, like when Asn turns to Asp, or IsoAsp residue(s). This certifies that deamidation occurs, and quantitatively increases if the enzyme is left at 90 °C for a long time, in line with the data reported by Zale and Klibanov [34]. Again, the bands fading augments with time, indicating an actual massive deamidation triggering RNase A precipitation, as we detected, although under different conditions, in [32]. Incidentally, we recall that Seshadri and colleagues found that RNase A loses half of its helix plus two third of β-sheet contents when it is denatured at 65 °C, although in that case at pH 2 [40]. Therefore, we can envisage that the acid environment compensates the 65 to 90 °C ΔT to cause denaturation irreversibility.

### 3.6. Stability of RNase A Dimers in Solutions Containing Urea

Since the T_m_ of monomers and dimers were almost the same (Table 1), the thermal denaturation of the RNase A species is not affected by the presence or absence of quaternary structure. Therefore, we measured the stability of the dimers in the presence of a chemical denaturant. To do so, we incubated for 2 h at 25 °C N_D-H/Et_ or C_D-H/Et_ samples dissolved in 20 mM NaPi, pH 6.7 at 0.5 to 0.7 mg/mL concentration, in the presence of urea concentrated up to 6 M. The cathodic PAGE analysis of the samples performed under non denaturing conditions revealed the quaternary structure of both N- and C-dimers resisted almost totally up to 4 M urea (not shown), and only a partial dissociation occurred with 5 M urea (Figure 3, left panel). Then, while in 6 M urea N-dimers are still partially present in dimeric form (not shown), C-dimers totally dissociate to monomer (Figure 3, right panel). Considering the high urea concentration, we can envisage the surface area forming the quaternary structure might partially protect dimers from the destabilizing effect of the denaturant.

### 3.7. Intrinsic Fluorescence Spectra Suggest Slight Structural Modifications Only within the Differently Produced RNase A Dimers

The intrinsic fluorescence emission spectra of RNase A X_-H_ and X_-Et_ species were recorded in 20 mM NaPi, pH 6.7, by exciting each 0.1 mg/mL protein solution at 280 nm. Both M_-H_ and M_-Et_ exhibit fluorescence emission profiles with a 305 nm maximum, identical to the one of native M (Figure 4A). Then, N_D-H_ and C_D-H_ shared similar emission spectra with one another, with an emission maximum centered at about 307 nm, while both N_D-Et_ and C_D-Et_ exhibited spectra characterized by a broader band displaying emission maxima with a red-shift of 3 nm with respect of those of N_D-H_ and C_D-H_ (Figure 4B).

Such a variation may suggest a not negligible difference in the affection occurring in both N_D-Et_ and C_D-Et_ tertiary structures with respect to the influence induced by acid lyophilization, this effect occurring around one or more aromatic residue(s).

### 3.8. The Enzymatic Activities of RNase A Species vs. ss- or ds-RNA Are Partly Affected by the Incubation Applied to Oligomerize the Enzyme

The enzymatic activity of all RNase A monomers and dimers vs. yeast ss-RNA was spectrophotometrically measured at 300 nm, as indicated by Kunitz [35]. The relative kinetic parameters are reported in Table 1, and the curves visible in Figure 5A,B indicate that yeast RNA is more easily attacked by the monomers than by N_D-H/Et_, and even more than by C_D-H/Et_, similarly to previously reported data [2,28]. However, we deduce now from the K_M_ values that the affinity to yeast RNA of M_-H_ and M_-Et_ is lower than the one of native M (Table 1, Figure 5A). Then, C_D-H_ displayed a higher V_max_ than C_D-Et_, while only negligible differences emerged within N_D-H_ and N_D-Et_ (Table 1, Figure 5B).

The activity of RNase A vs. ds-RNA is known to be low [1], but also to increase with the enzyme self-association, either proportionally to oligomers size or to their basic charge density exposure [2,5,28,41]. Hence, as it is here for both N_D_s vs. C_D_s couples (Figure 5C), the RNase A oligomers exposing different basic charge density (Figure 1A, Appendix A) provide significantly different activity values, in line with previously reported data [2,5,28,30,38]. To compare the effects induced by alternative treatments, we spectrophotometrically measured at 260 nm the activity of all X_-H_ and X_-Et_ species vs. increasing concentrations of the synthetic poly(A):poly(U) ds-RNA substrate (Figure 5C). However, we stopped the analysis when Abs_260_ reached a too high photometric error, and we therefore investigated only the initial linear part of the Michaelis-Menten curves: M, M_-H_ and M_-Et_ displayed the same activity values within one another, as well as N_D-H_ with respect to N_D-Et_. Instead, C_D-H_ was slightly more active than C_D-Et_, as it can be envisaged also from the data reported in Table 1.

### 3.9. Anti-Tumor Activity of RNase A Dimers on Human MeWo and A375 Melanoma Cell Lines

We previously reported that RNase A oligomers display a cytostatic effect in acute myeloid leukemia cell lines and reduce also tumor growth derived from human melanoma cells transplantation in nude mice [29]. In order to compare here the anti-tumor activity of RNase A dimers, we incubated human MeWo and A375 melanoma cells with 25, 50, 100 µg/mL of N- or C-dimers obtained by applying the two aforementioned HAc- or EtOH-methods on RNase A.

Preliminary results (not shown) indicated the anti-tumor activity of the dimeric species suffers time latency, similarly to that exerted by ONC in A375 cell line [42,43]. Therefore, we report in Figure 6A,B the data of cell viability assays measured with sulforhodamine B (SRB) sodium salt after 72 h cells incubation with all RNase A dimers.

Assays were paralleled by incubations with 100 µg/mL M_-H_, or M_-Et_. Indeed, both monomers were not able to inhibit MeWo or A375 cell growth and represent negative controls [25]. Similarly, in MeWo cells, no significant reduction of cell viability occurred after 72 h incubation with 25 and 50 µg/mL C_D-H_ or C_D-Et_ species, whereas a significant cell viability decrease occurred only at the highest concentration (100 µg/mL) of C_D-Et_, but not of C_D-H_. Conversely, both N_D-H_ and N_D-Et_ exerted a concentration-dependent reduction of MeWo cell viability reaching, at 100 mg/mL concentration, about 50% cell viability reduction with respect to the untreated control (100% viability), or to each corresponding, inactive M_-H_ or M_-Et_ (Figure 6A). On the other hand, the A375 cell viability was indeed significantly affected by both dimeric species regardless to the method used for their oligomerization, except for C_D-H_ and N_D-H_ species administered at the lowest concentration of 25 µg/mL (Figure 6B).

## 4. Discussion

The RNase A self-association occurring through the 3D-DS mechanism and the mutations favoring this event have been deeply studied in the recent past [2,4,6,8,9,10,12,24,32,44]. Notably, 3D-DS is shared by several amyloidogenic proteins [17,18,44] that form toxic vs. non-toxic oligomers as a function of the environmental conditions [20,21,22,23]. Notably, despite being native RNase A not amyloidogenic [24], its oligomers can exert a suitable antitumor activity [25].

Therefore, we firstly investigated here if different conditions inducing RNase A oligomerization [3,13] may affect the structural, enzymatic, as well as biological properties of the N- and C-swapped dimers or of the residual monomers. Besides, also MS, cation-exchange chromatography, and cathodic non denaturing PAGE (Appendix A, Figure 1A) showed no deamidation affecting all X_-H_ and X_-Et_ species depending to the incubation protocol applied [3,13]. Thereafter, a DVS cross-linking reaction [33] stated here for the first time that, besides N_D-H_ and C_D-H_ [14,15], also N_D-Et_ and C_D-Et_ form through 3D-DS (Figure 1B). Then, the far UV-CD spectra indicate that both treatments do not significantly affect either the secondary structure of all RNase A species (Figure 2A). The near-UV-CD spectra, registered only for the monomers because of yield limitations [13], indicate that also the tertiary structures are not affected (Figure 2B). However, lacking RNase A Trp residues, its near UV-CD profile is contributed by only Phe and Tyr that in turn are not evenly distributed in the structure. Hence, possible local and slight conformational changes might not be excluded.

The T_m_ values of both RNase A M_-H_ and M_-Et_, as well as of all dimers, fall very close to the 65 °C value of native M (Table 1). Instead, harsh 90 °C incubation allows a complete RNase A renaturation only if this condition is maintained for less than ten minutes (Figure 2C,D), in agreement with other authors reports [34,39,40]. By contrast, the 60 °C incubation in 40% aqueous EtOH, as well as the 40% HAc lyophilization, induce only a slight RNase A destabilization [2,3,13]. Consequently, advantageous condition(s) for a successful RNase A self-association should result from a compromise between harsh and weak denaturing environments. In line with this argument, a 70 °C incubation RNase A in 40% EtOH was previously found to provide a lower yield of oligomeric species than at 60 °C [13,34].

Then, chemical denaturation with urea did not show a differential behavior of the RNase A species as a function of the method used for their formation [3,13]. Instead, the although partial resistance of both N- and C-dimers in urea up to 5 M (Figure 3, left panel) is somehow surprising, considering either their non-covalent nature or that it was previously found that urea does not promote RNase A oligomerization [15]. The open interfaces [7] of the N- and C-dimers [14,15], not present in the monomer [7], and the stabilizing effect of phosphate linking the two active site His 12 and 119 swapped residues [45,46] could however justify this result. Then, dimers resistance diminishes to zero for C_D-H/Et_ in 6 M urea (Figure 3, right panel). Indeed, RNase A is known to have just started its unfolding in 6 M urea [47], and this could suggest that dimers dissociation is almost contemporary to enzyme unfolding.

The intrinsic fluorescence emission spectra did not show significant differences within native RNase A (M), M_-H_ and M_-Et_ (Figure 4A). Instead, both N_D-Et_ and C_D-Et_ spectra are significantly different from the corresponding N_D-H_ and C_D-H_ (Figure 4B), suggesting different conformational changes deriving from thermal incubations with respect to acid lyophilization. Variations may affect regions containing Phe and Tyr residues, but their extent is difficult to be evaluated if we consider that N_D_ displays an intrinsically different structure from C_D_ [14,15]. Therefore, despite fluorescence variations (Figure 4), the enzymatic activity of the two N_D_ (Table 1, Figure 5) could be less affected than C_D-H_ and/or C_D-Et_ by different treatments for their known intrinsic higher compactness and lower flexibility than the latter ones [48,49].

The K_M_ values of both M_-H_ and M_-Et_ measured vs. yeast RNA double the one of native M (Table 1), indicating the enzyme affinity is reduced upon recovered from harsh incubations (again, Figure 1, Appendix A). The N_D-H_ activity vs. yeast RNA is about the same of N_D-Et_, but both species display V_Max_ values being around half the native monomer. Instead, either C_D-H_ or C_D-Et_ are less active than N-dimers, and C_D-Et_ in turn even less than C_D-H_ (Table 1, Figure 5B). Similar M vs. N_D_/C_D_ activity differences had been reported in the past [28], but we now substantiate them with kinetic parameters. The lower activity displayed by RNase A-N_D_ than the monomer had been previously justified in terms of a minor number of available cutting active sites per each dimer than per monomer alongside the ss-RNA substrate filament [28]. Table 1 reports that both N- and C-dimers share similar K_M_ values vs. ss-RNA, indicating a quite similar affinity for the substrate. Conversely, they display different V_Max_ values: therefore, when N_D_ is saturated by a ss-RNA substrate, its turnover number is about twice the one of C_D_. To explain this difference, we advance two hypotheses: (i) although N_D_ and C_D_ active sites should be almost identical, since 3D-DS warrants the restoring of the so-called functional unit [4], they belong to protein scaffolds endowed with different conformation and flexibility [48,49], that would account for the N_D_ vs. C_D_ different V_Max_ values registered; (ii) considering the two active site clefts as being reciprocally “cis-like-oriented” in RNase A-N_D_ [14], while “trans-like-oriented” in C_D_ [15] (Figure 7), we may envisage a single ss-RNA molecule saturates both active sites of only N_D_. Instead, the two C_D_ active sites saturation would require the concomitant and less likely binding of two ss-RNA molecules without affecting K_M_ but halving the turnover number of this dimer.

No significant differences have been instead registered within the activity of the three monomers measured vs. poly(A):poly(U) ds-RNA substrate, as well as within the two N-dimers, while C_D-Et_ is slightly less active than C_D-H_ (Figure 5C), as it occurs vs. ss-RNA (Table 1, Figure 5B). Although the kinetic parameters vs. this ds-RNA substrate could not be calculated, differences rely only within C-dimers. Indeed, the C-dimer is known to be intrinsically, about 4/5-fold or 2.5/3-fold more active vs. poly(A):poly(U) than RNase A monomer or N-dimer, respectively [2,5,28]. These differences had been ascribed not only to dimericity, as proposed by Opitz et al. [50], but also to the ability of an increasingly basic charge exposure (M < N_D_ < C_D_) [28] to induce the ds-RNA unwinding and ease ss-RNA tracts hydrolysis [41]. In this scenario, the fluorescence data may suggest again an alteration of the geometry of the C-dimers active sites, or of the catalytic subsites [51,52] regions, resulting different from EtOH thermal incubations rather than from HAc lyophilization. These differences may result larger within the two more flexible C_D-H_ or C_D-Et_ than within the two more rigid N-dimers [48,49].

Remarkably, RNases can be considered non-mutagenic antitumor compounds for their ability to digest intracellular RNAs [53]. Nevertheless, monomeric RNase A can be cytotoxic only at very high concentrations [54], therefore exceeding the 1:1 RNase:RI stoichiometry [55]. Considering the high RI cytosolic concentration (about 4 µM) [56], one crucial event favoring RNases cytotoxicity is RI evasion. This is what natively monomeric ONC is intrinsically able to do, [26,57], and this allows it to be particularly active, as recently found also against melanoma cells [42,43]. In addition, natural dimeric bovine seminal (BS)-RNase [58,59] or bacterial dimeric binase [60] or also RNase A cross-linked dimers and trimers showed to be cytotoxic [61,62], differently from their monomeric counterparts [63]. Although 3D-DS N- and C-dimers, as well as larger oligomers did not show so far to affect the viability of solid tumors cells [64], they were active against cell lines from leukemic malignancies [29]. These data may suggest the anti-tumor effects are related to RNase(s) self-association, thus leading to an at least partial RI evasion of the oligomers. By the way, it was reported that an artificial RNase A tandem-dimer displayed a 1:1 stoichiometry with RI, despite having one active site per each subunit [53].

We report here the activity exerted by the differently produced RNase A N_D_ and C_D_ against human MeWo and A375 melanoma cell lines, and we compare these data with those previously obtained in an in vivo melanoma model [29]. We found no reduction of MeWo cell viability after 72 h incubation with C_D-H_, while a slight decrease was detectable only with C_D-Et_ 100 µg/mL, i.e., its highest concentration tested. Conversely, both N_D-H_ and N_D-Et_ reduced MeWo cell viability in a concentration-dependent manner (Figure 6A). Indeed, both N_D-H/Et_ retain a larger potential against MeWo melanoma cells viability than both C_D_ species. In that case, we can envisage that the higher compactness and lower flexibility of N_D-H/Et_ with respect to C_D-H/Et_ [48,49] favors N_D_ cell internalization and/or their routing in the cytosol, where they can exert cytotoxicity. On the other hand, the A375 cell viability was affected by both C_D_ and N_D_ couples (Figure 6B), suggesting the internalization and/or routing of RNase A dimers could be an event at least partly depending on the cell type treated. Interestingly, the EtOH thermal incubations seem to produce RNase A dimers exerting a slightly higher anti-tumor effect than the ones deriving from 40% HAc lyophilization against both cell types. In particular, the significant MeWo cell viability reduction exerted by 100 µg/mL C_D-Et_ but not by C_D-H_ deserves attention. Again, the significantly higher effect produced by both 25 µg/mL C_D-Et_ and N_D-Et_ on A375 cells viability than the same concentrated species obtained through HAc lyophilization has to be underlined.

Finally, we recall that the results obtained here with two melanoma cell lines parallel previous data collected in vivo against human melanoma cells transplanted in nude mice: indeed, both RNase A dimers (in that case, only N_D-H_ and C_D-H_ were tested) inhibited the tumor growth, even if N_D-H_ was slightly more effective than C_D-H_ [29].

## 5. Conclusions

The data collected here indicate both acid lyophilization and thermal incubation protocols [3,13] produce the same RNase A 3D-DS oligomeric species, although a high oligomers yield is favored by the former method [3] (Appendix A).

Then, the spectroscopic and kinetic data, the relative parameters, the hypotheses advanced to explain the higher catalytic activity of N_D-H/Et_ than C_D-H/Et_ vs. ss-RNA, the behavior of N_D-H/Et_ and C_D-H/Et_ in urea, as well as the analysis of their anti-tumor activity certainly represent interesting novelties for these species.

In any case, the significant anti-tumor activity exerted by the N-dimers in both human MeWo and A375 melanoma cell lines, as well as the activity of C-dimers vs. A375 cells, can represent a seed to develop new non mutagenic therapies to counteract with pt-RNases tumors that are presently still incurable [65].

## Figures and Tables

**Figure 1 life-11-00168-f001:**
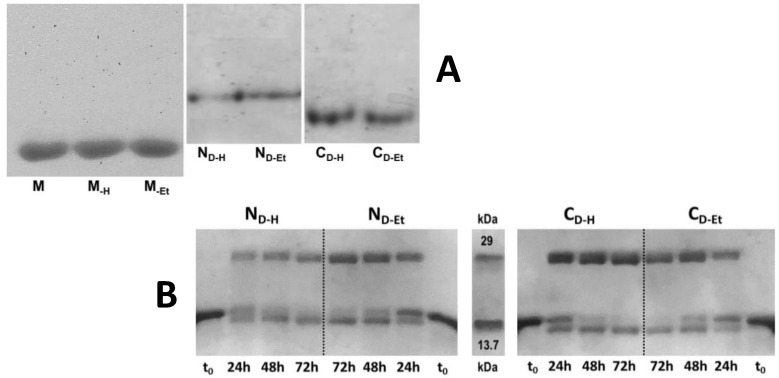
Analyses of the RNase A species recovered after different incubations types. (**A**) Non denaturing cathodic PAGE of RNase A monomers (M_-H_, M_-Et_ and native M), of N-dimers (N_D-H_, N_D-Et_) or C-dimers (C_D-H_, C_D-Et_), recovered after lyophilizing the enzyme from 40% HAc solutions or after its 2 h 40% EtOH incubation at 60 °C. Each 5–6 μg sample was electrophoresed for 90 min in a 15% polyacrylamide gel in the sample buffer tank immersed in an ice bath [30]; (**B**) 15% polyacrylamide SDS-PAGE of N_D-H/Et_ (left panel) and C_D-H/Et_ (right panel), after DVS cross-linking obtained by adding 10% v/v DVS in a 1000 molar excess over each 0.5 mg/mL protein sample incubated in a 0.1 M NaAc/HAc buffer, pH 5.0 [33]; 10 µg aliquots of each reaction mixture were collected and treated every 24 h with β-mercaptoethanol to stop the reaction, and about 6 µg were injected in the gel. MW standards: RNase A, 13.7 kDa; carbonic anhydrase, 29 kDa.

**Figure 2 life-11-00168-f002:**
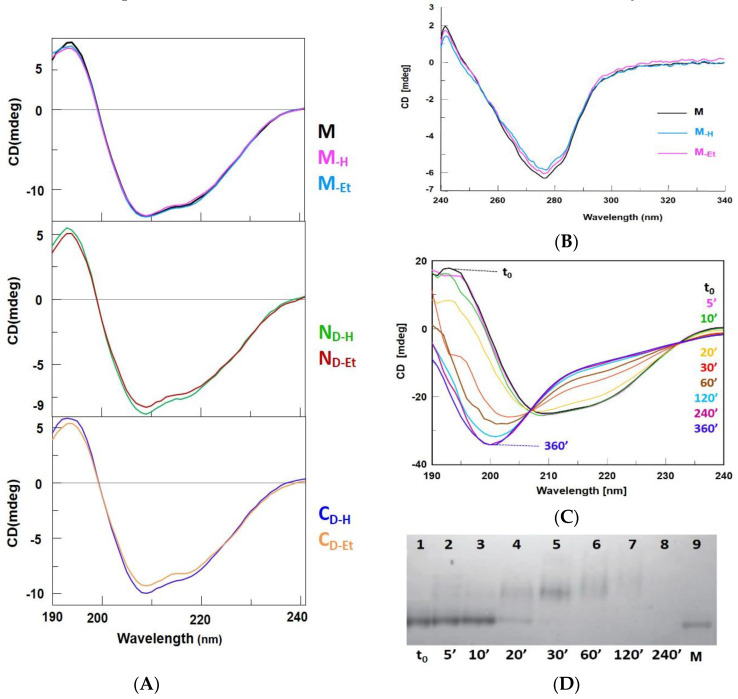
CD spectra and thermal denaturation of RNase A monomers. (**A**) Far UV-CD spectra of 0.30 mg/mL RNase A monomers (upper panel: native M, black; M_-H_, magenta; M_-Et_, cyan), and of 0.10 mg/mL N-dimers (middle panel: N_D-H_, green; N_D-Et_, red), or C-dimers (lower panel: C_D-H_, blue; C_D-Et_, orange) dissolved in 10 mM NaPi, pH 6.7, recorded from 190 to 240 nm. Each curve represents three different overlapping measurements recorded per each RNase A species. (**B**) Near UV-CD spectra of native RNase A M (black line), M_-H_ (magenta), M_-Et_ (cyan). Spectra were recorded from 240 to 340 nm in 10 mM NaPi, pH 6.7. Sample concentrations were 0.45–0.50 mg/mL each. Curves are representative of three different overlapping measurements recorded with each monomer. (**C**) Far UV-CD spectra time-course of RNase A native monomer after its incubation at 90 °C. Times indicated in the panel: t_0_, black; 5′, pink; 10′, green; 20′, yellow; 30′, red; 60′, brown; 120′, cyan; 240′, violet; 480′, blue. Curves are representative of three different experiments showing the same trend; (**D**) Cathodic PAGE analysis of samples collected from the same time-course reported in panel C. In lane 9, M: native, untreated, RNase A, used as a standard.

**Figure 3 life-11-00168-f003:**
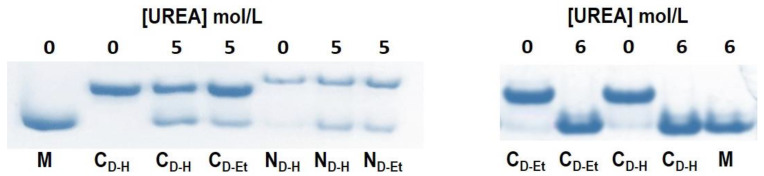
Non denaturing cathodic PAGE analysis of N- and/or C-dimers (-H/-Et) after incubation with 5 M (**left panel**) or 6 M (**right panel**) final concentration urea. Dimers dissolved in 20 mM NaPi, pH 6.7 were brought to about 0.5–0.7 mg/mL with the indicated urea concentrations and incubated for 2 h at 25 °C. Then, about 5 to 7 μg samples were electrophoresed for 90 min in a 12.5% polyacrylamide gel, with the sample buffer tank immersed in an ice bath [30].

**Figure 4 life-11-00168-f004:**
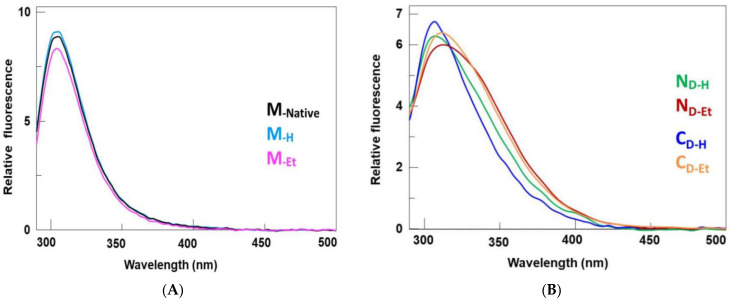
Intrinsic fluorescence emission spectra of (**A**) native RNase A monomer, M (black line), M_-H_ (magenta), or M_-Et_ (cyan), and (**B**) N_D-H_ (green), N_D-Et_ (red), C_D-H_ (blue) or C_D-Et_ (orange). Spectra were registered upon exciting at 280 nm a 0.1 mg/mL solution of each RNase A species dissolved in 20 mM NaPi, pH 6.7. Curves are representative of three measurements displaying almost identical results and performed with three samples for each species.

**Figure 5 life-11-00168-f005:**
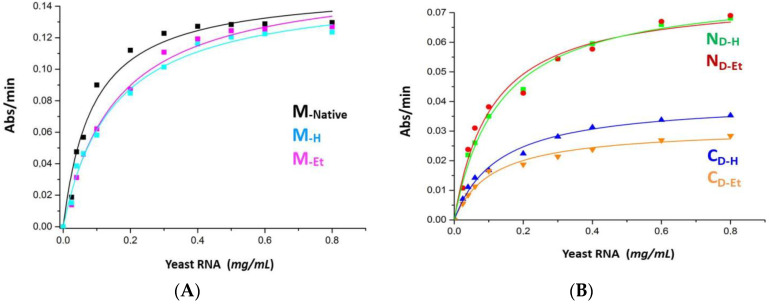
Enzymatic activity vs. ss- and ds-RNA of RNase A monomers and dimers. Kunitz activity [35] curves of (**A**) RNase A monomers (R^2^ values: M 0.978; M_-H_ 0.991; M_-Et_ 0.991), and (**B**) N- and C-dimers (R^2^ values: N_D-Et_ 0.976; N_D-H_ 0.993; C_D-Et_ 0.990; C_D-H_ 0.991), vs. yeast RNA (ss-RNA) and as a function of substrate concentration. Color codes are the same of Figure 4. Species were dissolved in 20 mM NaPi buffer, pH 6.7, at a concentration spectrophotometrically determined with ε^1%^_280_ = 7.3 [31]. Amounts of enzyme added to the substrate: monomers, 0.25 µg of a 0.5 mg/mL solution; dimers, 0.4 μg of 0.4–0.5 mg/mL solutions. Each value is the average of triplicates, and RNA activities are expressed as Abs/μg enzyme*min [35]. The relative kinetic parameters are reported in Table 1; (**C**) Initial, linear part of the enzymatic activity curves of the RNase A species as a function of increasing concentrations of the ds-RNA poly(A):poly(U) substrate [36] measured up to the limit dictated by an excessive Abs_260_ spectrophotometric error. Amounts of 20 mM NaPi pH 6.7 enzyme solutions added to the substrate: 10 μg of each 1 mg/mL monomer; 4 μg N_D-H/Et_, or 2 μg C_D-H/Et_, respectively, of 0.4–0.5 mg/mL solutions [5]. Each reaction was followed at 260 nm in the linearity range, while values reported in the curves are the mean of quadruplicates ± S.D., with specific activities expressed as Abs/μg enzyme*min [36].

**Figure 6 life-11-00168-f006:**
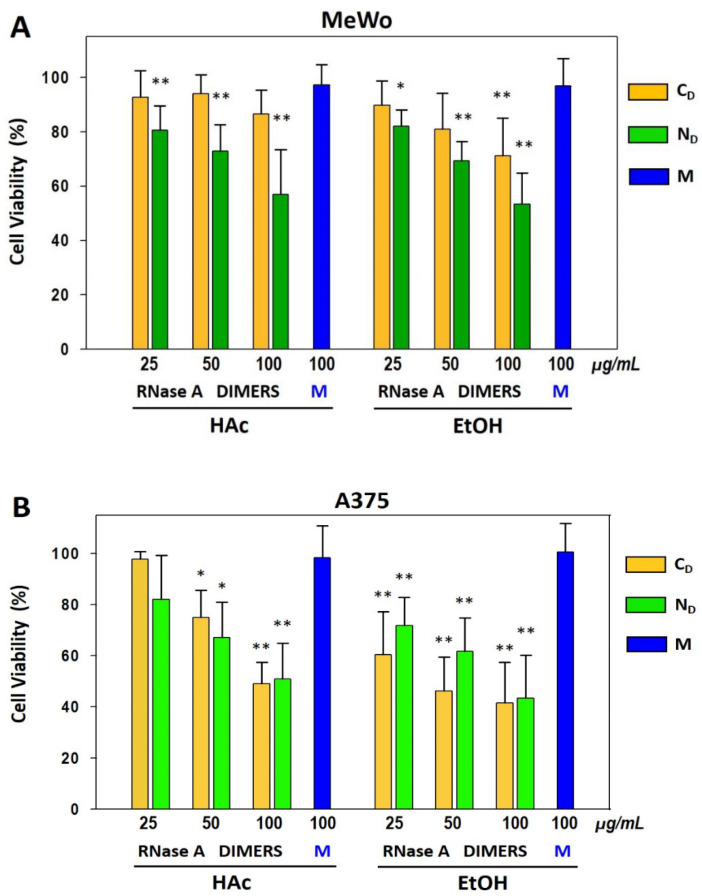
Effects of RNase A monomers or of N- or C-swapped dimers on (**A**) MeWo and (**B**) A375 human melanoma cell viability. Cells were treated with the indicated concentrations of the different RNase A species, as reported in Mat. & Met. After 72 h, SRB cell viability assays were performed. Bars represent the percentage with respect to negative controls of cell viability measured after incubating cells with 25, 50, or 100 µg/mL RNase A N_D_ (green bars), C_D_ (yellow bars), or with 100 µg/mL monomers (M, blue, negative controls). RNase A species were recovered from 40% HAc lyophilization (left bars group) [3], or upon 2 h incubation at 60 °C in 40% EtOH (right bars group) [13]. All values reported are the mean ± S.D. of four to five independent experiments, each performed in six replicates. The statistically significant differences in cell viability induced by the dimeric species vs. each relative monomer are shown (* *p* < 0.05; ** *p* < 0.01).

**Figure 7 life-11-00168-f007:**
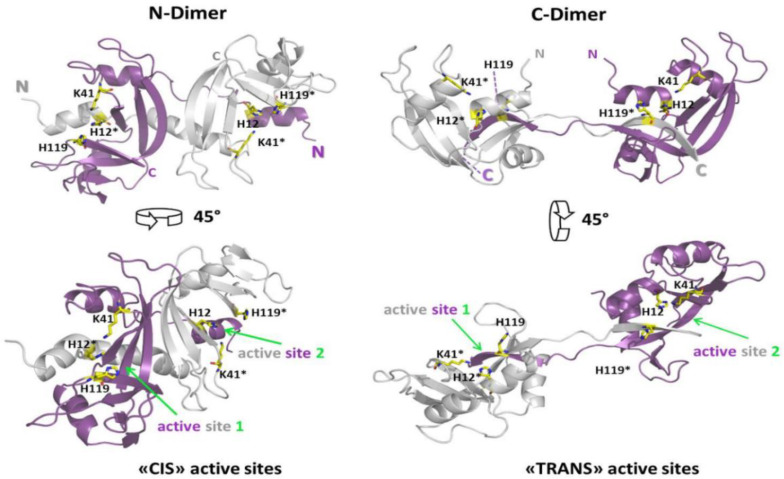
Relative orientation of the two active sites in the two domain-swapped RNase A dimers. Ribbon representations of the RNase A N-dimer (1A2W.pdb [14], **left**) and C-dimer (1F0V.pdb [15], **right**). The subunits of each dimer subunit are colored in grey or purple, respectively, and the active site residues, His 12, His 119, and Lys 41 are indicated, as well as in the upper panel the positions of N- and C-termini. The asterisks***** denote the residues of the grey subunit, to better highlight the domain restoring the composite active site upon domain swapping [7]. In lower panels, the “CIS-like” or “TRANS-like” reciprocal orientations of the active sites are better visible by a 45° rotation. Images are rendered with the PyMol software (Schrödinger).

**Table 1 life-11-00168-t001:** T_m_ values and kinetic parameters of RNase A species.

		Yeast RNA (ss-RNA)		Poly(A):Poly(U) (ds-RNA)
RNaseA Species	T_m_ (°C)	V_max_ (ΔAbs_300_/min/µg)	K_M_ (mg/mL)	Spec.Activity ^1^/[RNA] (mg/mL)
M (native)	64.9 ± 0.1	0.151 ± 0.006	0.087 ± 0.012	13.0 ± 0.3
M_-H_	65.0 ± 0.1	0.152 ± 0.005	0.147 ± 0.015	13.1 ± 0.2
M_-Et_	64.7 ± 0.1	0.160 ± 0.005	0.156 ± 0.016	13.4 ± 0.3
N_D-H_	64.6 ± 0.1	0.079 ± 0.002	0.131 ± 0.012	25.8 ± 0.5
N_D-Et_	64.2 ± 0.1	0.076 ± 0.004	0.112 ± 0.018	24.1 ± 0.4
C_D-H_	65.0 ± 0.1	0.041 ± 0.002	0.130 ± 0.012	57.5 ± 0.6
C_D-Et_	65.2 ± 0.1	0.030 ± 0.001	0.109 ± 0.011	48.5 ± 0.7

^1^ Calculated as ΔAbs_260_/min/µg enzyme [30]. The data reported in this column are the slope, i.e., Specific activity/[RNA] derived from the curves reported in Figure 4C.

## Data Availability

Commercial human melanoma cell lines.: MeWo (HTB-65™) and A375 (CRL-1619™) cell lines (ATCC^®^, Manassas, VA, USA).

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
