# Peer review of "RNase A Domain-Swapped Dimers Produced Through Different Methods: Structure–Catalytic Properties and Antitumor Activity"

_life, 2021, doi:10.3390/life11020168_

Round 1

Reviewer 1 Report

In this work, the authors studied RNase A oligomerization and in detail structural and/or functional properties associtaed to two different protocols tested. Data and strategies are well presented and discussed.

Some minor comments for the paper are mentioned below:

  • Stability of RNase A species should also be evaluated in the presence of chemical denaturants;
  • Toxicity on tumoral cells would require at least one other cell line;
  • Discussion paragraph is too long and speculative;

Moreover, there are some typographical errors, or venial inaccuracies, to correct:

Ex:

  • In paragraph 2.11, page 5, please add "o" to "tw";
  • In the legend of figure 5, lyophlization must be replaced with lyophilization;
  • Verbs in the present and past tense are often used in the same paragraphs (see paragraphs 2.6, 3.5
  • in the legends of figures 3 and 4, pH values of the buffers should be added
  • In the title of paragraph 3.9, "on human melanoma" is more appropriate than "in human melanoma" 

Reviewer 2 Report

The ability of RNase A to oligomerize through domain swapping has been one of the most intriguing phenomena discovered in the early years of structural biology.  Since then, this aspect has been the subject  by a remarkable number of studies. In this scenario, Gotte and coworkers report a comparative analysis of the biophysical and functional properties of domain-swapped RNase A dimers obtained using different experimental protocols. Considering that the topic is of interest, that the experiments are technically sound and that the manuscript is clearly written, I personally believe that this work can be considered for publication. I would suggest a mention either in the Introduction or in the Discussion to the RNaseA mutants that favor oligomerization.

Minor point

In line 243 “tw “  should be “two”
